Evidence for protection of targeted reef fish on the largest marine reserve in the Caribbean

Pina-Amargós Fabián 1
González-Sansón Gaspar 2
Martín-Blanco Félix 3
Valdivia Abel 4 abel.valdivia@unc.edu
1 Centro de Investigaciones de Ecosistemas Costeros , Cayo Coco , Morón , Ciego de Ávila , Cuba
2 Department of Studies for Sustainable Development of the Coastal Zone, University of Guadalajara , Jalisco , Mexico
3 Florida Fish and Wildlife Conservation Commission, Fish and Wildlife Research Institute , Tequesta , FL , USA
4 Department of Biology, University of North Carolina at Chapel Hill , Chapel Hill , NC , USA
Elphick Chris
Electronic publication date: 2014 Feb 20
Publication date: 2014
Volume: 2
Electronic Location ID: e274
Received 2013 Oct 4; Accepted 2014 Jan 25
Copyright: © 2014 Pina-Amagós et al.
Copyright year: 2014
Copyright holder: Pina-Amagós et al.
License: This is an open access article distributed under the terms of the Creative Commons Attribution License, which permits unrestricted use, distribution, and reproduction in any medium, provided the original author and source are credited.
License URL: https://creativecommons.org/licenses/by/3.0/

Keywords: Marine reserves, Coral reefs, Overfishing, Target reef fish

Funding: Funding was provided by the Ministry of Science, Technology and Environment of Cuba to Fabian Pina-Amargós. The funders had no role in study design, data collection and analysis, decision to publish, or preparation of the manuscript.

==============================
Marine reserves can restore fish abundance and diversity in areas impacted by overfishing, but the effectiveness of reserves in developing countries where resources for enforcement are limited, have seldom been evaluated. Here we assess whether the establishment in 1996 of the largest marine reserve in the Caribbean, Gardens of the Queen in Cuba, has had a positive effect on the abundance of commercially valuable reef fish species in relation to neighboring unprotected areas. We surveyed 25 sites, including two reef habitats (reef crest and reef slope), inside and outside the marine reserve, on five different months, and over a one-and-a-half year period. Densities of the ten most frequent, highly targeted, and relatively large fish species showed a significant variability across the archipelago for both reef habitats that depended on the month of survey. These ten species showed a tendency towards higher abundance inside the reserve in both reef habitats for most months during the study. Average fish densities pooled by protection level, however, showed that five out of these ten species were at least two-fold significantly higher inside than outside the reserve at one or both reef habitats. Supporting evidence from previously published studies in the area indicates that habitat complexity and major benthic communities were similar inside and outside the reserve, while fishing pressure appeared to be homogeneous across the archipelago before reserve establishment. Although poaching may occur within the reserve, especially at the boundaries, effective protection from fishing was the most plausible explanation for the patterns observed.

Introduction

Marine reserves have been largely beneficial for the recovery of fish density, biomass, and diversity (Côté, Mosqueira & Reynolds, 2001; Halpern, 2003; Lester et al., 2009; Molloy, McLean & Côté, 2009). Studies have shown an increase in abundance of targeted species, families, and even functional groups after the establishment of no-take marine reserves (Russ, Alcala & Maypa, 2003; Alcala et al., 2005; Claudet et al., 2008). Positive effects have been observed in average size (20–30% increase), species richness (11–23% increase), and reproductive capacity (Mosquera et al., 2000; Côté, Mosqueira & Reynolds, 2001; Russ & Alcala, 2003; Palumbi, 2004). The benefits of protection have been detected as early as one to five years following fishing bans (Gell & Roberts, 2003; Halpern, 2003; Russ, Alcala & Maypa, 2003; Palumbi, 2004) with positive effects increasing over time (Halpern & Warner, 2002; Maypa et al., 2002; Alcala et al., 2005; Claudet et al., 2008). Although marine reserves are presumed to protect several species from exploitation, not all species respond positively to protection (Claudet et al., 2010).

The response to protection is greatly variable among fish taxa depending on their commercial value, body size, mobility, and other life-history traits. Overall, strongly exploited species of larger body size tend to respond significantly better and faster than unexploited and relatively smaller species (Mosquera et al., 2000; Russ, Alcala & Maypa, 2003; Claudet et al., 2008, 2010). Furthermore, relatively long-lived species with high mobility and variable recruitment may respond more slowly to fishing closures than short-lived species with narrow spatial requirements and steady recruitment (Gell & Roberts, 2003; Russ, Alcala & Maypa, 2003; Palumbi, 2004). In fact, beneficial effects could take decades to detect in very mobile species. For instance, pelagic fish species with movement patterns that extend beyond reserve boundaries, respond slower than less vagile coastal species (Roberts & Sargant, 2002; Micheliet al., 2004). Nonetheless, exploited mobile species with wide home ranges may still benefit from protection (Claudet et al., 2010). In contrast, non-commercial bycatch and unexploited species rarely respond to protection and may even show declines after fishing has ended due to different life-history and ecological traits such as body size, habitat preferences and schooling behavior (Palumbi, 2004; Claudet et al., 2010).

Several factors independent of life history traits can also hinder the detection of positive effects in marine reserves. Dissimilarities in habitat structural complexity and benthic community composition can drive differences in fish assemblages that are not related to protection status, as the abundance of several fish species is correlated with substratum characteristics (McClanahan, 1994; Roberts & Sargant, 2002; Friedlander et al., 2003; Harborne, Mumby & Ferrari, 2012). Pre-exiting spatial patterns in fish abundance can influence species-specific response in marine reserves (Karnauskas et al., 2011). Similarly, the acquired behavior of targeted fish species towards divers (e.g., attraction due to feeding practices in protected areas, or avoidance due to spearfishing in non-protected areas) could lead to overestimation or underestimation of fish abundances (Kulbicki, 1998; Hawkins et al., 1999). Additionally, lack of appropriate spatial and temporal replication in some studies (Halpern & Warner, 2002; Halpern, 2003; Alcala et al., 2005), the use of different methods to compare fish abundances (Maypa et al., 2002; Ward-Paige, Mills Flemming & Lotze, 2010), temporal fluctuations in population abundance (Babcock et al., 2010), and ineffective enforcement (Pomeroy et al., 2005; Mora et al., 2006) can thwart the detection of beneficial effects of marine reserves.

Most studies addressing the effects of marine reserves on fish assemblages in the Caribbean have focused on relatively small protected areas (Polunin & Roberts, 1993; Roberts, 1995; Roberts & Hawkins, 1997; Roberts et al., 2001). This is because few relatively large and continuous marine reserves exist in the region and their fish communities can be highly variable due to natural intra-habitat differences (Chiappone & Sullivan-Sealey, 2000; Harborne et al., 2008). Large and especially older marine reserves, however, may have more implications for the recovery of large and mobile reef fish populations than smaller reserves at scales necessary for conservation and fisheries management (Halpern, 2003; Claudet et al., 2008; Gaines et al., 2010). But effective enforcement and management in large marine reserves is difficult to achieve, especially in developing countries where there are limited resources for conservation (Mora et al., 2006; Guidetti et al., 2008). Thus, understanding the dynamics of relatively large marine reserves in protecting fish populations where enforcement may be a problem will provide crucial insights into reserve design and management needs.

Here for the first time, we analyzed the spatial and temporal differences of the density of relatively large-bodied and commercially important reef fish species inside and outside the largest marine reserve of the Caribbean, the Gardens of the Queen (“Jardines de la Reina”) national park in Cuba, established in 1996 (Appeldoorn & Lindeman, 2002). Several reef sites in the park are known to support some of the highest levels of fish biomass in the entire region (Newman et al., 2006). However, no comprehensive study has analyzed the effectiveness of this protected area in enhancing reef fish populations. Our study focused on the response of fish species that were historically targeted in the region before reserve establishment and that are still targeted outside the reserve (Pina-Amargós, González-Sansón & Cabrera-Paez, 2008a). We hypothesized that fish densities of these species are significantly higher inside the reserve than in neighboring unprotected areas due to protection from fishing. We discuss whether differences were independent of moderating factors such as habitat heterogeneity or the structure of benthic communities (coral and algae assemblages). Furthermore, we analyzed whether fish density differences were related to differential protection levels, fishing pressure before protection, alteration of fish behavior, or variation in spatial recruitment across the park. Our study provides useful information about the implications that large marine reserves have in developing countries with very limited resources for appropriate enforcement and effective reserve management.

Material and methods

Study sites and survey design

The Jardines de la Reina archipelago (hereafter JDLR) consists of ∼661 keys and covers ∼360 km in south-central Cuba (Fig. 1). In 1996, approximately 950 km2 of the archipelago, including a variety of coral reef, seagrass and mangrove systems, was proclaimed by the Cuban Ministry of Fisheries as a “zone under special regime of use and protection”. This management category is equivalent to the internationally recognized “Marine Reserve” terminology that will be used in this manuscript. This park is considered the largest continuous marine reserve in the Caribbean (Appeldoorn & Lindeman, 2002) with an area more than twice that of the Exuma Cays Land and Sea Park (442 km2) in the Bahamas (Chiappone & Sullivan-Sealey, 2000).

Figure 1 Location of survey sites across the Gardens of the Queen (Jardines de la Reina) archipelago.

Solid black circles are sites where reef slope and reef crest were surveyed. White circles represent sites where only reef slope was sampled. Dashed line delimits the marine reserve established in 1996. Solid lines divide the archipelago into five zones. NRW, Non-Reserve West; RW, Reserve West; RC, Reserve Center; RE, Reserve East; NRE, Non-Reserve East. For location coordinates refer to Table S1.

There are no quantitative data describing the reef fish and benthic community structure before reserve establishment. Previous studies indicated, however, that catch and fishing pressure were homogeneously distributed across the entire JDLR archipelago before protection, suggesting similar abundance of economically valuable species across the region (Pina-Amargós, González-Sansón & Cabrera-Paez, 2008a; Claro et al., 2009). Although after the declaration of the reserve fishing efforts were relocated to outside the reserve, poaching is still present towards the boundaries of the protected area (Claro, Lindeman & Parenti, 2001; Pina-Amargós, González-Sansón & Cabrera-Paez, 2008a). In fact, due to limited park resources and the extensive area to cover there is a gradient of effective protection from the center of the reserve (with higher enforcement due to a research station) to the boundaries of the reserve (with lower enforcement) that may affect fish communities (Pina-Amargós, González-Sansón & Cabrera-Paez, 2008a).

We estimated the spatial and temporal differences in density of relatively large and commercially valuable reef fish species inside and outside of the JDLR marine reserve. To analyze inter-habitat variability, we sampled two distinct reef habitats; reef slope (depths 8–15 m) and reef crest (depths 1–3 m). We accounted for location effects by surveying sites at both ends of the marine reserve. To stratify our survey, we divided the study area into five zones (Fig. 1), identified as Non-Reserve West (NRW), Reserve West (RW), Reserve Center (RC), Reserve East (RE) and Non-Reserve East (NRE). For reef slope habitats, we sampled 15 sites within the reserve (five sites equidistant in each of the three reserve zones) and 10 sites outside the reserve (five sites in each of the two non-reserve zones) (Fig. 1, Table S1). Reef crest habitats were only surveyed in NRW, RW and RC because the reef crest in RE and NRE were shorter in length (<500 m) than was required for our survey method (see below). Thus, for reef crests we surveyed eight sites within the reserve (four sites in RW and four sites in RC) and four sites outside the reserve in NRW. To access seasonal differences, we surveyed all sites five times, during June of 2004 and January, April, September, and December of 2005. Based on Pina-Amargós, González-Sansón & Cabrera-Paez (2008a), reserve enforcement follows this pattern by zones RC > RW > RE > NRW > NRE, where RC had high protection, RW and RE moderate protection, and NRW and NRE no protection.

To design this study, we used fish density means and variances from a pilot survey to estimate effect sizes and mean squared error. We performed an a priori power analysis for two and three-way ANOVAs with different sample sizes (e.g., 2 vs. 3 transects per site, 4 vs. 5 sites per zone) and numbers of factors (e.g., sites × zones × time vs. zones × time). We found that the analyses with two factors (5 zones and 5 times), with two transects per site, and five sites per zone, showed a power of at least 80%, indicating relatively high power for our study design. We did not include habitat as a factor nested within sites because the model was not balanced. This was because the two habitats did not occur in every site or because habitats differed from typical standard reef slope and crest (e.g., patchy reef track). During the previous pilot study, each permanent belt transect was marked with bottom buoys, underwater reference points were photographed, and their coordinates were registered using a GPS unit.

Fish densities

Underwater visual censuses for large mobile reef fish were used for fish counts (methods modified from Richards et al., 2011). That is, instead of towed divers, swimming divers performed the surveys. We randomly deployed two permanent belt transects (800 × 10 m) at each slope site and two (500 × 10 m) at each crest site. Shorter transects were used on the latter because the minimum continuous reef crest track found was 500 m. During each survey, divers counted individual fish in a 10 × 10 m area in front of them for ∼1–2 min moving sequentially along the transect when all fish were recorded. This approach was useful to avoid recording the same fish more than once, ensuring a near instantaneous sampling design, and minimizing changes in fish behavior due to diver presence (Ward-Paige, Mills Flemming & Lotze, 2010). Body size (fork length in centimeters, FL) of each individual was estimated in 10 cm intervals, as recommended by Westera, Lavery & Hyndes (2003). Each transect was surveyed in ∼40–50 min. Before beginning every survey the divers became acquainted with the belt transects’ width (10 m) using a metric tape.

For the surveys, we selected only 28 reef fish species of high commercial value that are often targeted by fishermen (Claro et al., 2009) (Table 1 and Table S2). We chose these species based on information obtained from semi-structured interviews of local fishermen to determine the most common targeted fish species and their minimum catch size (“trophy size”). Semi-structured interviews consisted of a limited and formal set of questions, but new questions were added as a result of what fishermen said. The results of the interviews indicated that these 28 relatively large species (Table 1) were of high commercial value and the most targeted by local fishermen (F Pina-Amargós, unpublished data). The term “trophy species” will be used for these targeted species in this paper.

Table 1 Overall descriptive statistics of targeted trophy fish species by reef habitat to determine frequent species for the analysis.

Trophy size, average density (number of individuals per 1000 m2 ± 1 standard error) above trophy size, and entire body-size range are shown. Trophy size for each species was determined based on semi-structured interviews and was defined as the minimum fish size that fishermen would catch for that species. Frequency (f%) was defined as the proportion of sites within the reef habitat at which individuals of the species above trophy size occurred. Sample size was 250 transects (5 months × 5 zones × 5 sites × 2 transects) for reef slope and 120 (5 months × 3 zones × 4 sites × 2 transects) for reef crest. For taxonomic information of each species see Table S2.

		Reef slope	Reef crest	
Common name	Trophy (cm)	f (%)	Mean ± SE (ind. ∕1000 m2)	Size (cm)	f (%)	Mean ± SE (ind. ∕1000 m2)	Size (cm)	
Nassau grouper	55	96	0.48 ± 0.03	15–85	58	0.15 ± 0.02	15–65	
Hogfish	45	100	1.96 ± 0.08	10–65	84	0.63 ± 0.06	10–55	
Schoolmaster	35	100	17.57 ± 0.84	10–55	100	53.17 ± 2.16	10–55	
Cubera snapper	65	95	0.53 ± 0.05	25–125	66	0.22 ± 0.03	25–85	
Dog snapper	55	87	0.40 ± 0.05	15–85	97	0.78 ± 0.06	15–85	
Mutton snapper	45	94	0.38 ± 0.03	15–75	82	1.06 ± 0.15	15–65	
Yellowfin grouper	55	94	0.47 ± 0.03	15–75	69	0.25 ± 0.03	15–65	
Tiger grouper	55	96	0.47 ± 0.03	15–75	89	0.35 ± 0.04	15–75	
Black grouper	65	83	0.24 ± 0.02	15–105	79	0.24 ± 0.02	15–95	
Great barracuda	85	94	0.31 ± 0.02	35–135	78	0.31 ± 0.03	25–105	
Spotted eagle ray	150	9	0.03 ± 0.002	105–235	2	0.01 ± 0.001	95–165	
Yellow jack	55	14	0.11 ± 0.05	35–75	20	0.06 ± 0.001	25–75	
Crevalle jack	55	11	0.39 ± 0.04	35–85	5	0.12 ± 0.01	25–75	
Horse-Eye jack	55	37	0.43 ± 0.02	25–75	26	0.28 ± 0.03	25–85	
Reef shark	150	3	0.01 ± 0.001	95–205	10	0.03 ± 0.003	105–215	
Silky shark	150	14	0.03 ± 0.001	135–255	0	—	—	
Southern stingray	150	35	0.10 ± 0.03	65–175	21	0.07 ± 0.002	75–135	
Goliath grouper	75	15	0.05 ± 0.003	65–255	4	0.03 ± 0.001	55–135	
Nurse shark	150	43	0.10 ± 0.02	75–255	36	0.12 ± 0.01	85–205	
Margate	45	23	0.06 ± 0.002	25–65	0	—	—	
Tarpon	100	19	0.36 ± 0.03	95–205	26	0.16 ± 0.02	85–195	
Lemon shark	150	0	—	—	10	0.03 ± 0.002	155–205	
Rainbow parrotfish	55	18	0.05 ± 0.002	45–115	37	0.12 ± 0.02	45–115	
Midnight parrotfish	55	5	0.04 ± 0.002	35–95	19	0.08 ± 0.003	35–105	
King mackerel	75	9	0.03 ± 0.002	65–135	0	—	—	
Spanish mackerel	55	11	0.03 ± 0.001	55–95	0	—	—	
Cero	55	42	0.13 ± 0.02	25–65	7	0.03 ± 0.001	25–55	
Permit	55	4	0.01 ± 0.000	35–75	8	0.01 ± 0.001	45–85	

For all comparative analyses among zones and time, we selected the most frequent fish species (f > 50%) from the 28 trophy species surveyed across sites (Table 1). Frequency (f) was defined as the proportion of all surveys within a given reef habitat on which a given species was detected. Species with frequencies <50% were not included in the analyses because the power of detecting differences among reserve and non-reserve sites with our study design was relatively low and the results could lead to misleading conclusions. This was based on the results of the power analysis, which suggested that more than two transects were needed per site to compare relatively low frequency species meaningfully. Additionally, to increase the probability of detecting differences due to fishing, we only used individuals larger than the species-specific trophy size in the analyses (Table 1). This approach made comparisons between non-reserve and reserve sites more meaningful as fishermen mostly target individuals above the trophy size. Of the most frequent species, we analyzed the data including and excluding the schoolmaster (Lutjanus apodus). This species shows strong schooling behavior (Claro, Lindeman & Parenti, 2001), was the most abundant in most sites, and is the least commercially attractive based on the semi-structured interviews.

Spatial and temporal differences of fish biomass among reserve and non-reserve sites were not analyzed in this study and are beyond the scope of our objectives. This was because we were only interested in analyzing differences in fish densities of the most targeted and commercially valuable species above a certain trophy size to determine the effectiveness of reserve protection.

Reef structural complexity and benthic communities

We did not include reef structural complexity or benthic community characteristics as cofactors in the models. A previous study characterized in detail the reef architectural complexity and benthic community structure (mainly corals and algae) of the reef sites analyzed in this study during the same time period (Pina-Amargós et al., 2008c; Table S3). That study found no significant differences in reef structural complexity, corals, or algal assemblages among reserve and non-reserve sites within the same reef habitat. Specifically, most of the values of substrate rugosity, coral cover, density of coral colonies, coral bleaching prevalence, coral mortality percentage, density of coral recruits, algae cover (divided into six functional groups) were statistically similar across all zones within the same reef habitat independent of protection status (Pina-Amargós et al., 2008c). It is unlikely that relatively large reef fish species respond to small changes in benthic community species composition (coral and algae). Instead, coarse variables such as reef rugosity, total coral cover, algae cover, or number of coral colonies seems to be more important (Wilson, Graham & Polunin, 2007; Harborne, Mumby & Ferrari, 2012). Therefore, if these habitat variables are similar across zones (within habitat type) as reported by Pina-Amargós et al. (2008c), it is unlikely that they will drive any differences in the spatial density distribution of trophy fish species among zones. For detailed information refer to Table S3.

Data analysis

Statistically significant differences in mean density were assessed using a two-factorial fixed-effects analysis of variance (two-way ANOVA), considering levels of protection (five zones) and sampling time (five months) as factors. We used the combination of transects and sites as replicates within the zones to increase power in the analysis. We tested the assumptions for the ANOVA using the Shapiro-Wilk test for normality and Levene’s test for homogeneity, following the criteria suggested by Underwood (1996) and Quinn & Keough (2002). When these assumptions were not met, transformations were required to resolve violations (Table S4). To test for independence of the model residuals we examined spatial autocorrelation among zones within habitats for each trophy fish species using Moran’s I similarity spline correlograms (Bjørnstad & Falck, 2001). Spatial autocorrelation for the crest habitat among three zones was not calculated because at least four zones are required for the analyses (Fig. S1).Temporal autocorrelation among months was analyzed using the autocorrelation function from the package stats in R (Figs. S2 & S3). There was no significant spatial or temporal autocorrelation for any of the trophy species among zones or months within habitats, supporting the assumption that the residuals of the ANOVA model were independent of each other (Figs. S1, S2 & S3). For the two-way ANOVA, the F and P values of the interaction effects are presented in Table 2. When the interaction effects were not significant the statistical results of the main effects are presented within the text. Habitat structural complexity and benthic community composition were not included in the models because no differences were found in these factors among reserve and non-reserve sites within the same reef habitat (Pina-Amargós et al., 2008c). For graphical representation of the significant interactions we constructed bubble scatterplots, where the circle diameter is proportional to the mean density of trophy fish in each combination of zone and sampling time. Using a Welch’s t test (i.e., modified Student’s t test for two samples with unequal variances (Ruxton, 2006)), we also analyzed the differences between protection levels based on the pooled mean density for each trophy species. This latter analysis provides strong evidence that differences in mean density of trophy species between reserve and non-reserve sites are present even after combining the variability detected in space and time. Data were analyzed using the software STATISTICA 8.0 (StatSoft, 2007). For autocorrelation analyses we used the package ncf 1.1–4 and stats in R v3.0.1 (R Core Team, 2013).

Table 2 Summary statistics from the factorial ANOVAs and Welch’s t-tests performed on density data for the ten most frequently occurring species (f > 50%).

(A) F-ratios and p-values are for the interaction term (zone × time) within habitats for the two-factorial ANOVA. Degrees of freedom for the interaction and residual are in parenthesis. (B) Values of Welch’s t and p-values are for the comparison between reserve and non-reserve sites. For the ANOVA, only the results of the interactions are shown for brevity, see main text for significant main effects.

A. ANOVA	Reef slope × Time	Reef crest × time	
Species/groups	F (16, 215)	p	F (8,105)	p	
Black grouper	4.05	<0.001a	1.09	0.378	
Yellowfin grouper	2.38	0.003a	5.52	<0.001a	
Tiger grouper	2.54	0.001a	1.18	0.321	
Schoolmaster	2.24	0.005a	1.69	0.109	
Nassau grouper	3.37	<0.001a	2.14	0.038a	
Cubera snapper	1.95	0.018a	3.54	0.001a	
Dog snapper	3.52	<0.001a	3.52	0.001a	
Mutton snapper	3.38	<0.001a	2.37	0.022a	
Hogfish	2.08	0.010a	2.36	0.023a	
Great barracuda	2.20	0.006a	0.67	0.716	
Total trophy	25.67	<0.001a	12.61	<0.001a	
Total trophy (no schoolmaster)	20.81	<0.001a	7.55	<0.001a	
B. Welch’s t-test	Reef Slope	Reef Crest	
Species/groups	t Welch	p	t Welch	p	
Black grouper	2.23	0.027a	1.33	0.188	
Yellowfin grouper	1.28	0.201	2.11	0.037a	
Tiger grouper	0.48	0.632	0.46	0.648	
Schoolmaster	0.24	0.804	0.38	0.706	
Nassau grouper	0.96	0.340	0.79	0.429	
Cubera snapper	0.85	0.393	2.26	0.026a	
Dog snapper	0.79	0.429	0.82	0.415	
Mutton snapper	2.71	0.007a	3.26	0.001a	
Hogfish	2.96	0.003a	3.49	0.001a	
Great barracuda	1.41	0.159	0.83	0.372	
Total trophy	0.18	0.854	0.14	0.886	
Total trophy (no schoolmaster)	1.44	0.151	2.06	0.041a	
a at a level of <0.05

Results

The JDLR archipelago showed a relatively high frequency and density of commercially valuable fish species. Of the 28 fish species surveyed, 10 were present in at least 50% of all transects and were categorized as frequent (Table 1). Of this group, schoolmaster was the most frequent and abundant species in both reef habitats (slope and crest), with mean densities one or two orders of magnitude higher than the rest of the species (Table 1). Schoolmaster was also three times more abundant on reef crests than on reef slopes due to the schooling behavior of the species. Overall, these 10 species, except dog snapper, were more frequent on the slope than on the reef crest (Table 1). Hogfish and mutton snapper had the second highest densities on reef slopes and reef crests respectively. Frequent trophy species showed a range body size of 10–135 cm and 10–105 cm fork length (FL) in reef slope and reef crest habitats respectively (Table 1). Body size range for cubera snapper, black grouper and great barracuda (e.g., 15–135 cm FL) were at the higher end of this size range, while hogfish and schoolmaster (e.g., 10–65 cm FL) occupied the lower end (Table 1). Trophy size of frequent species fell slightly above the middle point of their body size range found during surveys (Table 1).

The factorial analysis of variance within habitats indicated that on reef slopes the interaction between reef zones and time was significant for all of the 10 most frequent trophy species (Table 2A, Fig. 2). This indicates that the spatial distribution patterns in average density of these species varied across zones in the archipelago depending on the time of the survey (Fig. 2). For example, dog snapper had the highest densities in January in RC, but by September the highest density was found outside the reserve in NRE (Fig. 2). Although we found a great degree of variability among trophy species, for most of them the highest average density per zone tended to be in September, while December appeared to show the lowest values (Fig. 2). Overall, with few exceptions, all these species showed a trend towards higher densities inside rather than outside the marine reserve during the study, especially in RC (Fig. 2). The pooled mean densities within the reef slope habitat and by protection level (combining all transects during the survey) showed the same trend, however, only the densities of three out of ten species (i.e., mutton snapper, black grouper, and hogfish) were significantly higher inside than outside the reserve (Fig. 3A, Table 2B). Mutton snapper and black grouper showed a two-fold increase, while hogfish had a 1.7-fold increase from non-reserve to reserve (Fig. 3A). The rest of the species did not show overall differences between protection levels. Within the reserve, schoolmaster and hogfish had the highest densities, with the former having one or two orders of magnitude higher than the rest of the species (Fig. 3A).

Figure 2 Comparisons of relative mean densities of targeted trophy species (above trophy size) on reef slope habitats for each zone and survey time.

Circle diameters are proportional to the mean density of each species at each combination of surveyed site and time. NRW, Non-Reserve West; RW, Reserve West; RC, Reserve Center; RE, Reserve East; NRE, Non-Reserve East. Survey date labels show month (first two letters) and year (last two digits). For hogfish, circle diameters are half size (×0.5) due to proportionally higher mean densities than the rest of the species.

Figure 3 Differences in fish densities between reserves and non-reserve sites for targeted trophy species.

Pooled mean densities (number of individuals/1000 m2± 95% confidence interval) for targeted trophy species on reef slopes (A) and reef crest (B) by protection level. Non-reserve sites (white bars) and reserve sites (gray bars). Horizontal arrows denote significant differences (Table 2, Welch’s t-test, * p < 0.05, ** p < 0.01, *** p < 0.001).

The analysis of variance for the reef crests showed that six out of the ten most frequent fish species (i.e., mutton snapper, cubera snapper, dog snapper, Nassau grouper, yellowfin grouper and hogfish) showed significant interactions between zones and time (Fig. 4, Table 2A). This analysis also indicates that fish densities within reef crest habitats varied spatially during the study. Overall, these six species tended to have higher densities inside than outside the reserve during the study, especially in RW (Fig. 4). In contrast, the density of the rest of the species (i.e., black grouper, tiger grouper, schoolmaster, and great barracuda) showed no interactions among zones and time. Density of black grouper differed among zones with higher values inside than outside the reserve (F2,105 = 6.35, p = 0.002), but showed no difference among months. Tiger grouper and schoolmaster showed no spatial and temporal variation in densities (Table 2A), while great barracuda only showed seasonality (F4,105 = 3.24, p = 0.015) but no difference among zones (Table 2A). As with reef slopes, the pooled mean density within the reef crest showed a trend towards higher densities of trophy species inside the reserve (Fig. 3B). Yet, only four out of the ten most frequent trophy species (i.e., mutton snapper, cubera snapper, yellowfin grouper, and hogfish) had statistically significant differences (Fig. 3B, Table 2B). From non-reserve to reserve within the reef crest, mutton snapper showed an average eight-fold increase, cubera snapper and hogfish ∼4-fold, while yellowfin grouper had ∼3-fold increase in density. Within the same habitat, schoolmaster had the highest density, one order of magnitude higher than the other species; mutton snapper was next most numerous (Fig. 3B).

Figure 4 Comparisons of relative mean densities of targeted trophy species (above trophy size) on reef crest habitats for each zone and survey time for the significant interactions from the factorial ANOVA.

Circle diameters are proportional to the mean density of each species at each combination of survey site and time. NRW, Non-Reserve West; RW, Reserve West; RC, Reserve Center. Survey date labels show month (first two letters) and year (last two digits). For mutton snapper, circle diameters are half size (×0.5) due to proportionally higher mean densities than the rest of the species.

Finally, densities of the ten most frequent trophy species were combined as a group (i.e., trophy species density) and significant interactions between zones and time were detected on both reef slopes and the reef crest, both when including or excluding schoolmaster (Table 2). Overall, higher densities of all trophy species combined were found inside the reserve than outside (Fig. 5). On reef slopes the trend was similar with and without schoolmaster. In this habitat, the highest density of trophy species was found in Jun 2004 in RW followed by Jan 2005 in RW and RC. In contrast, on reef crests, total trophy density was higher for Jul 2004, Jan 2005 and Apr 2005 in RC, but by Sep 2005 higher densities were observed in RW and NRW. By Dec 2005, the three reef crest zones had comparable total densities of trophy species (Fig. 5). However, the density of trophy species in reef crest zones when schoolmaster was excluded from the analysis, tended to be higher inside than outside the marine reserve over time (Fig. 5). This result indicated that on the reef crest schoolmasters had a strong effect on the total trophy species density among zones and time (Table 2).

Figure 5 Comparison of pooled density averages for the most frequent (>50%) trophy species by reef habitat for each zone and survey time.

(A) includes the first ten species in Table 1. (B) includes nine species but excludes the schoolmasters. Circle sizes are proportional to the mean density of each group. NRW, Non-Reserve West; RW, Reserve West; RC, Reserve Center; RE, Reserve East; NRE, Non-Reserve East. Survey date labels show month (first two letters) and year (last two digits). For reef crest with schoolmaster, circle diameters are half size (×0.5) due to proportionally higher densities. For ANOVA results refer to Table 2A.

Discussion

Our results support the hypothesis that the implementation of the JDLR marine reserve has promoted significantly higher densities of some commercially valuable and relatively large reef fish species. This result is consistent with previous meta-analyses that found that the greatest benefits of reserves are the recovery of exploited species, especially large ones (Côté, Mosqueira & Reynolds, 2001). The relatively large size of this reserve (over 900 km2) may have provided greater benefits to species, such as top predators, that have large area requirements and that are not effectively protected in small reserves (Halpern, 2003). Recovery rates of larger vagile fish species in our study cannot be directly estimated as we do not have information from before reserve establishment. An alternative approach would be to use the differences between reserve and non-reserve sites for those trophy species that had significant overall differences (e.g., mutton snapper, cubera snapper, black grouper, yellowfin grouper and hogfish). However, this information would be incomplete and misleading as we only analyzed above trophy size individuals. Furthermore, there are no published studies of marine reserves in the Caribbean that use long transects (e.g., 500–800 m) as sampling unit to survey large vagile reef fish species. This is because most reserves in the region are relatively small and large fishes are low in numbers. Thus, any comparison of density values with other marine reserves could be inaccurate as we have only analyzed a subset of the population and used a different survey method.

There is no long-term data set addressing changes in the fish communities before and after this reserve establishment, therefore evidence of protection based on a before and after approach is impossible to demonstrate. Nonetheless, the patterns observed in the density of trophy species are not likely the response to confounding factors such as, reef type heterogeneity, spatially different fishing efforts before reserve establishment, lack of appropriate replication, alteration of fish behavior due to divers, differential recruitment, or a combination of them. By surveying multiple control sites outside the reserve, replicating our study in space and time, and critically analyzing some potential cofactors that were not included in the models, it seems likely that protection from fishing, and a gradient of enforcement from the center to the boundaries of the reserve, is the most plausible explanation for the patterns observed.

Habitat structural complexity and benthic community structure were not likely the drivers of the differences observed in trophy fish densities within the same habitat across sites in our study. Structural complexity is often a significant factor influencing reef fish assemblages in coral reefs (Sale, 1991; Harborne, Mumby & Ferrari, 2012). For instance, the three dimensional structure of corals can affect fish recruitment patterns (Sale, 1991), provide refuge by reducing predation risk (Hixon & Beets, 1993), and increase shelter in high-flow environments (Johansen, Bellwood & Fulton, 2008). There is in fact a long recognized positive correlation between coral cover and the abundance and diversity of reef fishes (Jones et al., 2004). Moreover, coral loss due to bleaching events has considerably altered the population dynamics of reef fish species that rely on live coral for food or shelter (Jones et al., 2004; Graham, 2007). The benthic structure and composition in reserve sites may foster more fish abundance, regardless of local protection. As noted however, a previous study showed no significant differences in reef structural complexity, benthic community composition, coral and algae cover, or bleaching prevalence among reserve and non-reserve sites within the same reef habitats (Pina-Amargós et al., 2008c) (see Table S3). Based on the homogeneity of the benthic community and reef structural complexity, the differences observed in fish assemblages among zones within habitat types are likely independent of these factors.

Understanding fishing pressure before the establishment of marine reserves is important to determine the potential effects of protection after fishing has ended (Russ & Alcala, 1998; Halpern, 2003; Alcala et al., 2005; Osenberg et al., 2006; Lester et al., 2009). This approach is fundamental to avoid confounding factors such as spatial differences in fishing activities. In JDLR, for both reef habitats, in all zones except NWR, there appear to be no differences in catch and fishing effort along the archipelago before reserve establishment, suggesting similar spatial abundance of finfishes before protection (Pina-Amargós, González-Sansón & Cabrera-Paez, 2008a; Claro et al., 2009). With the declaration of the marine reserve in 1996, catch and fishing effort were relocated to outside the reserve (Claro, Lindeman & Parenti, 2001; Pina-Amargós, González-Sansón & Cabrera-Paez, 2008a). After over ten years of protection, Pina-Amargós, González-Sansón & Cabrera-Paez (2008a) found a strong negative association between landings and fish abundance of most commercially important species across the JDLR archipelago. This result suggests that fishing pressure has been lower inside the reserve where fish were more abundant (Pina-Amargós, González-Sansón & Cabrera-Paez, 2008a). Thus, the fish abundance distribution in JDLR (greater inside the reserve) was not likely related to uneven fishing pressure before the establishment of the protected area.

Lack of appropriate replication or control sites in studies that detected the effects of marine reserves could also lead to misleading conclusions and unsound management policies (Willis et al., 2003). Our analysis was based on strong experimental design as recommended by other studies (Halpern, 2003; Willis et al., 2003). Our patterns were robust in both habitats and across the five sampling periods, thus we can state that the differences inside and outside the marine reserve persist at spatial and temporal scales.

Observations of fish behavior in the JDLR archipelago showed that species of most commercial value tended to flee from divers when closely approached more often in non-reserve sites than reserve sites (Pina-Amargós et al., 2008b). Comparisons of flight distance (i.e., distance at which an organism begins to flee an approaching threat) inside and outside long-established reserves indicate fish behavior can be modified by the presence/absence of fishing (Gotanda, Turgeon & Kramer, 2009; Feary et al., 2011). For example, fish respond to divers in fished areas by fleeing or swimming away, while in protected areas they are less afraid and more curious (Gotanda, Turgeon & Kramer, 2009; Feary et al., 2011). Thus, this modified behavior can be used as a metric of fishing intensity. The previously reported contrasting behavior of trophy species between outside and inside the JDLR (Pina-Amargós et al., 2008b) support our hypothesis of stronger protection in the reserve.

Differences in fish behavior towards divers inside and outside the reserve may have influenced our results. If divers were significantly altering fish behavior, and flight distance of trophy species was sufficiently greater outside than inside the marine reserve, we may have underestimated fish densities outside the reserve due to lower fish detectability where individual fishes are fleeing from observers (Gotanda, Turgeon & Kramer, 2009; Feary et al., 2011). Conversely, large fish can approach divers inside reserves where feeding activities are common, which may lead to overestimation of density values (Kulbicki, 1998; Hawkins et al., 1999). However, our sampling methods and speed of survey minimized the interaction between fish and divers, hence reducing the possibility of changes in fish behavior due to the diver presence (McClanahan et al., 2007; Ward-Paige, Mills Flemming & Lotze, 2010; Richards et al., 2011). Thus, it is unlikely that the differences in fish density between protected and not protected zones were influenced by the observers.

Potential net movement of adult fish out of the marine reserve could also be evidence of effective protection within the reserve. After years of protection, fish and larvae tend to migrate from areas of higher abundance (e.g., inside reserves) to areas of lower abundance (e.g., outside reserves). This pattern is also known as spillover effect and has been reported in several long-established and well-functioning protected areas (Russ & Alcala, 2003; Alcala et al., 2005; Francini-Filho & Moura, 2008; Halpern, Lester & Kellner, 2009). Spillover effects within the JDLR archipelago have been experimentally confirmed through density manipulation of large-size and commercially valuable reef fish species using tagging methods and visual census (Pina-Amargós et al., 2010). Although the study was performed at a relatively small scale, the authors found that net emigration rates of tagged fish were two-fold higher than at control sites when a strong fish density gradient was established after modifying fish abundance (Pina-Amargós et al., 2010). In addition, anecdotal accounts of spillover effects of adult fish from the JDLR reserve reported by fishers (e.g., “fish leave the reserve and for that reason we catch more fish now than we did before”) support the scientific findings. Thus, if the evidence of net movement of adult fish in the JDLR archipelago towards the exterior of the reserve is true, then protection might explain a fish density gradient.

Differential recruitment inside and outside the reserve is also unlikely at the spatial scale of our study and may not influence the differences observed. To our knowledge, only two studies have addressed fish larval transport in Cuba (Lindeman et al., 1999; Paris et al., 2005). Both studies modeled larval transport through simulations from spawning aggregation sites for grunts and snappers in the southwest region (Lindeman et al., 1999) and for five snapper species (all of them included in our study) around the Cuban shelf (Paris et al., 2005). In the simulations, Paris et al. (2005) included two spawning aggregation sites in or near JDLR and suggested that significant levels of self-recruitment (up to 80%) structure the snapper populations, especially in this region. Based on these studies, it is not possible to make strong inferences about the distribution of larval recruitment at the relatively small spatial scales of JDLR archipelago (∼350 km). However, the species in our study have monthly spawning aggregations (Lindeman et al., 2000; Claro & Lindeman, 2003; Claro et al., 2009) and several spawning aggregation sites for snapper and grouper have been reported in the southeast region of the island (Claro & Lindeman, 2003). In fact, a grouper spawning hot spot has been confirmed inside the reserve (in RC) (F Pina-Amargós, pers. obs.). Therefore, several potential spawning aggregation sites could be producing larvae that are being dispersed homogeneously along the entire JDLR archipelago.

Out of the ten trophy species analyzed in our study, at least six showed significantly higher densities in both reef habitats (slope and crest) inside the marine reserve at some time during the study. Differences between reserves and non-reserves may be stronger for the reef slope than the crest habitat where fishes are naturally more abundant. These species (yellowfin grouper, Nassau grouper, cubera snapper, dog snapper, mutton snapper and hogfish) are also among the most commercially valuable and targeted in the region (Claro & Lindeman, 2003; Pina-Amargós, González-Sansón & Cabrera-Paez, 2008a; Claro et al., 2009). Therefore, as expected, these species have benefited the most from protection when fishing stopped or was drastically reduced (Côté, Mosqueira & Reynolds, 2001; Micheliet al., 2004; Russ et al., 2008). Tiger grouper and schoolmaster did not clearly respond to protection and positive effects were not consistent between reef habitats for black grouper and great barracuda. These last four species are less targeted by recreational fisheries in the JDLR archipelago (Pina-Amargós, González-Sansón & Cabrera-Paez, 2008a; Claro et al., 2009). The main reason, based on the semi-structured interviews, was that schoolmaster was regarded as low quality for consumption in the region, which supports prior findings that less targeted species are generally unaffected by reserve establishment (Micheliet al., 2004; Claudet et al., 2010). Similarly, tiger grouper, black grouper and great barracuda are prone to ciguatera fish poisoning (i.e., fish that are toxic for human consumption due to accumulation of ciguatoxin) in the region and fishers may avoid them (Claro, Lindeman & Parenti, 2001). Fishing regulations in Cuba have limited catch allowances for larger specimens of black grouper, cubera snapper and dog snapper around the island (Claro et al., 2009) and although they are also prone to ciguatera (Claro, Lindeman & Parenti, 2001) they may still be caught and consumed by illegal fishing. Nonetheless, our results support the view that commercially valuable species have increased in abundance after the establishment of marine reserves, hence responding better to protection.

Finally, effective management is essential for the success of marine reserves (Pomeroy et al., 2005; Mora et al., 2006). Ultimately, the positive response of fish to protection is indicative of good compliance with fishing restrictions (Smith, Zhang & Coleman, 2006; Guidetti et al., 2008). Fish responses to protection can be indirectly used to evaluate the effectiveness of strict no-take areas (Smith, Zhang & Coleman, 2006; Guidetti et al., 2008). According to Pina-Amargós, González-Sansón & Cabrera-Paez (2008a), effective protection decreases from RC, with the least human impact, to RW and RE with moderate protection, and NRW and NRE with the highest human activity. Although the JDLR marine reserve is not formally enforced by any national entity (Pina-Amargós, González-Sansón & Cabrera-Paez, 2008a), the area has mostly escaped the high fishing pressure recorded in the rest of the Caribbean (Hawkins & Roberts, 2004). This difference seems related to the relative remoteness of the archipelago, the economic situation of the country, restricted accessibility to the park, and the limited resources (e.g., boats, fuel, ice) that local recreational and commercial fishermen have faced for decades (Claro et al., 2009). Enforcement in the park is achieved indirectly by the reserve users. For example, former fishermen have become tour operators for the small resort that operates within the park (within the RC zone), where only ~1000 divers and fly-fishers (catch and release) are allowed every year (Figueredo-Martín et al., 2010). This model has indirectly promoted a reduction of illegal fishing by fostering protection (Pina-Amargós, González-Sansón & Cabrera-Paez, 2008a) since the revenue from local tourism is much more profitable than fishing (Figueredo-Martín et al., 2010). Inadequate protection close to the boundaries of the JDLR park might be an issue (as reflected in relative lower trophy fish densities) because most of the tourist activities, and thus indirect enforcement, occur at the center of the reserve (F Pina-Amargós, pers. obs.).

In summary, our study supports the findings that large Caribbean reserves can work and effectively restore populations of highly valued fish species on different reef habitats. The density of six out of ten highly targeted and frequent species in the JDLR archipelago were higher inside than outside the marine reserve in both reef slope and reef crest during a one-and-a-half year study. Although poaching may occur within the reserve, especially at the boundaries, effective protection from fishing was the most plausible explanation for the patterns observed. Relatively large marine reserves in the Caribbean are necessary to ensure the protection of valuable fish species at scales necessary for conservation and fisheries management. The JDLR marine reserve is the largest in the region and could function as a source area for species that have been extensively depleted Caribbean wide.

Supplemental Information

Supplemental Information 1 Names and locations of study sites.

Names and coordinates (dd, mm, ss) of the study sites by zones and reef habitat type (slope and crest). Depth range for reef slopes was 12-15m while for reef crest was 2-3m. Sites without coordinate values were not sampled in that habitat.

Click here for additional data file.

Supplemental Information 2 Taxonomic information for the trophy species surveyed during the study.

Common names are organized in the same order as in Table 1.

Click here for additional data file.

Supplemental Information 3 Summary of the benthic habitat characteristics for each zone and reef type, modified from Pina-Amargös et al., (2008b).

Numbers are mean standard error. Time is the month and year of survey for algae assemblages. Superscript letters represent significant differences among zones within the same reef habitat after a one-way ANOVA and a Student-Newman-Keuls post hoc analysis (letters: p < 0.05, no letters: p > 0.05). No statistical tests were performed on % cover of algae functional groups.

Click here for additional data file.

Supplemental Information 4 Transformations required to meet the assumptions of the ANOVA.

Transformations are for each trophy species by reef habitat. Analyses were performed on the transformed data, figures are based on the raw data.

Click here for additional data file.

Supplemental Information 5 Spatial autocorrelation of the model residuals for trophy species on slope reef habitats.

Spline correlograms using Moran’s I similarity index and lag distance in kilometers showing the lack of spatial autocorrelation of the model residuals for each of the nine trophy species analyzed in the slope reef habitat. Lines are mean (middle line) and 95% confidence interval (outer lines). Horizontal line is zero correlation.

Click here for additional data file.

Supplemental Information 6 Temporal autocorrelation among months using the autocorrelation function for nine trophy species in reef slope habitats

Autocorrelation function (ACF) values showing the lack of temporal correlation among five months (lag) for the nine trophy species in the reef slope habitat that showed significant interactions between zones and times based on the factorial ANOVA.

Click here for additional data file.

Supplemental Information 7 Temporal autocorrelation among months for trophy species in the reef crest habitat.

Autocorrelation function (ACF) values showing the lack of temporal correlation among five months (lag) for six trophy species in the reef crest habitat that showed significant interactions between zones and times based on the factorial ANOVA.

Click here for additional data file.

The authors thank G Omegna (Pepe) and the workers of Azulmar for logistic support, E Sala for constructive comments, and R Ginsburg and P Kramer for helping to secure funding for two expeditions. We also thank the Ministry of Science, Technology and the Environment of Cuba for financial and logistical support, especially C Pazos Alberdi, R Gómez Fernández, A Zúñiga Ríos and R Estrada Estrada. Infinite thanks to the workers of the Centro de Investigaciones de Ecosistemas Costeros for their support in the field surveys, especially to W Acosta de la Red, A Zayas Fernádez, L Hernández Fernández, L Clero Alonso, T Figueredo Martín, P E Cardoso Gómez and V O Rodríguez Cárdenas. A research permit was acquired through the Centro de Inspección y Control Ambiental. We appreciate the reviews of L Carr, J Bruno and L Valdivia of early versions of this paper. We thank the Academic Editor for PeerJ, Chris Elphick, and Michelle Paddack, and an anonymous reviewer for their critical comments that have improved this manuscript.

Additional Information and Declarations

Competing Interests

Author Contributions

Authors declare that they have no competing interests, except that Academic Editor Dr. John Bruno is the PhD Adviser of Abel Valdivia. Fabián Pina-Amargós is an employee of Grupo de Investigación y Monitoreo de la Costa Sur de Cuba, Centro de Investigaciones de Ecosistemas Costeros.

Fabián Pina-Amargós conceived and designed the experiments, performed the experiments, analyzed the data, contributed reagents/materials/analysis tools, wrote the paper.

Gaspar González-Sansón conceived and designed the experiments, analyzed the data, contributed reagents/materials/analysis tools, wrote the paper.

Félix Martín-Blanco performed the experiments, contributed reagents/materials/analysis tools, wrote the paper.

Abel Valdivia analyzed the data, contributed reagents/materials/analysis tools, wrote the paper.

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
