# Peer review of "Evidence for protection of targeted reef fish on the largest marine reserve in the Caribbean"

_PeerJ, doi:10.7717/peerj.274_

## Round 0.1 · original submission · Major Revisions

Both reviewers agreed that this is an interesting topic and a well-executed study that warrants publication. I agree with their views, but would like to see their comments fully addressed, with point-by-point responses, in revision. I also have three general comments of my own.

1. The submitted manuscript contains many small grammatical errors. I have copy-edited the MS Word file to identify corrections (I will email this document separately). The authors should check that the suggested changes do not alter their meaning, and look out for other cases that I might have missed. I will review again for such items after revision. There are also a few internal style inconsistencies (e.g., in the references) that should be corrected.

2. The analytical approach employs simple ANOVAs and t-tests, rather than more complex multivariate mixed-modeling methods. The authors are clear that their goals are not to address the issues that the latter approach could allow them to investigate (e.g., spatial and temporal variation, effects of habitat features, etc.). This decision, however, means they need to be more careful about the claims that they make because the inferences that are possible are rather limited. I have commented on some specific examples on the edited MS Word file. There is also a risk that by not addressing spatial and temporal autocorrelation that the assumption that the residuals of the ANOVA models are independent of each other was violated. I would suggest doing the more detailed analysis, but at a minimum the authors need to provide diagnostic tests to demonstrate that model assumptions were met.

3. Finally, it appears that the authors have not conducted any corrections for multiple comparisons (i.e., to control the Type I error rate). Given that many of the tests are quite close to the P=0.05 level, this is a real problem and some of the conclusions could change as a result. If the corrections were done, then the authors should make that clearer. If not, some correction method should be used (and details provided on the method and decisions about, for example, how tests were grouped for correction). Alternatively, the shortcomings of inferences for P-values close to 0.05 should be discussed and some of the conclusions tempered accordingly.

Reviewer 1 ·

Basic reporting

The study employs standard methods of evaluating fish density inside and outside marine reserves, and the methods are very clearly and comprehensively described. You provide relevant information about the region, although I’d like to see the boundary of the reserve in Figure 1 (see below, General Comments to Authors). I also appreciated the citing of other literature pertaining to JDLR and the way these previous studies were used to support the reserve effect as opposed to differences based on habitat characteristics, etc. Although there are some grammatical errors, these will easily be addressed by a copy editor.

Experimental design

The experimental design is well explained and clearly laid out. However, I’d like to see more information about the “time” factor in the text. Although you can search for the months of sampling in the Figures, please also list these in the methods so that the reader is clear about which months correspond to time in the statistical model.
I also didn’t see any discussion of site as a factor in your model. Sites can sometimes have very different fish communities (even if they are similar in reef structural complexity, coral cover, algal cover as the data you cite demonstrate). Also, I’m curious about why you didn’t include habitats has a factor nested within sites. If this was simply because it wasn’t a balanced model (both habitats were not surveyed at all sites) then please state this, or if there was an ecological reason or no statistical significance, that should be stated as well.

Validity of the findings

This is a solid study of ecological responses inside a marine reserve, which is novel due to its focus on an important locality in the Caribbean. The consideration of the effect of enforcement on fish responses to reserve protection, and how it varies across different zones in the reserve, is also an important contribution to the MPA literature at large. The data are strong and the analyses appropriate (although see my comments on the experimental model, regarding the statistical model). For analyses such as these, it’s common to consider the other factors that could potentially lead to the differences in reserve and non-reserve fish densities. You carefully step through the data on habitat, behavioral, and recruitment-based scenarios that show the reserve effect is the primary driver of these significantly different densities.

Additional comments

Your analysis of the largest marine reserve in the Caribbean is valuable, not only because of the reserve’s size and unique location, but also because data from this area is hard to come by. I commend you for highlighting the previous work of Pina-Amargós and building upon it in this paper—these are data that are very relevant to the MPA community.

Specific points:

-Throughout: Just a note to remember to describe data in the plural form (i.e. using “data are” instead of “data is”)

-Line 12: Just FYI, Lester et al. 2009 is an update of the Halpern 2003 study you cite—it includes more sites and updated data.

-I’m curious as to whether you’ve done any gradient-based analysis on the non-reserve sites. NRE is particularly interesting to me, since the reserve boundaries subdivide what appears to be a relatively close group of smaller islands. I’d like to see that analysis—it would be very interesting to see whether the densities of fishes in Site 21 differ significantly from those at Site 25, for example.

-Figure 1: I find the dashed lines around each of the five zones confusing, primarily because my first thought is that these are reserve boundaries. But of course that is not true since the outermost zones are identified in name as the non-reserve sampling sites. It would be more intuitive to include the actual boundaries of the reserve as the outer dashed lines, use vertical dashed lines to subdivide the reserve into its RW, RC, and RE zones, and then leave the NRW and NRE zones open. For NRW and NRE it is already clear that they are divided from RE and RW and dashed lines around them is not necessary. This approach provides more information as well—now we can assess how far Sites 5 and 21 are from the reserve boundary.

-Figures 2, 5: Please define the (x0.5) in the legends.

·

Basic reporting

This manuscript is well structured and fairly clearly written. It is concise, with a clearly laid-out question that is well-justified by the introduction. Some restructuring would be helpful to increase clarity.
The abstract could use some more specific details including the year that the reserve was established, and the fact that the data were collected seasonally. The sentence “This trend was mostly consistent over time” is vaque. The last two sentences of the abstract should be dropped since they are not results from this study.
In the Introduction, the paragraphs beginning on line 23 and line 37 need to be re-organized. There are two different concepts presented that are co-mingled and should each have their own paragraph. Firstly, there are reasons why a reserve effect may not occur, which the 1st of these paragraphs includes, but the 1st sentence of the 2nd paragraph should be included with the prior paragraph since habitat differences can cause a null effect, rather than an effect that is there but not detected, which the 2nd paragraph should more cleanly address. Poor enforcement, noted in line 47 would be included in the 1st paragraph. Note also that there is evidence for both attraction of fish to divers and avoidance of them (in areas of spearfishing).
The age of the reserve is important and needs to be stated sooner – suggest in line 64.
Grammatical/language errors are as follows:
Line 33: after ‘home ranges’ should read “may still benefit..”
Line 34: bycatch misspelled
Line 58: should read “..where there are limited resources..”
Line 72: “discussed” should be changed to “discuss”. “modulating” should be “moderating”
Line 90: rather than “Few studies…”, state “Previous studies..” (few makes it sound like the support is poor, rather than the reality which is just that there are not many studies, but they find the same result). Otherwise, this paragraph is great support for the experimental design.
Line 120: change “latest” to “latter”
Line 136: be consistent with whether the interviews were “semi-structured” or “semi-structure” (the former makes more sense)
Line 174: should read “..composition were not included..”
Line 286: I think you mean “Understanding” rather than “understated”
Line 397: change “touristic” to “tourist”

Supplement Tables were not provided for the review.

Experimental design

This study has an excellent experimental design which incorporates boundary effects and separates habitats. The data are analyzed appropriately and all figures are clear and sufficient. Table 1 needs to be restructured so that the data are shown for inside vs. outside reserve, rather than as a single overall average, which is not ecologically relevant given the findings.
The justification for the study is excellent, however, I believe that the authors have missed an opportunity analytically that one expects from how they set up the introduction. Because there are so few large reserves, yet many publications of reserve effects, the authors have the opportunity to compare recovery rates of larger, more vagile species in this large vs other small reserves.
There could also be more clear analysis and discussion of the boundary effects, since this design is so well set up to observe them.

Validity of the findings

The conclusions are well-justified by the data and analysis.

---

## Round 0.2 · Minor Revisions

This is a nice revision and I have only minor editorial comments. I will send a marked up copy of the word file separately. Please check all of my suggested changes to ensure that none change your meaning. Please pay special attention to the numbering of figures and tables, which is confusing in a few places. If there should be additional supplemental figures please make sure they are uploaded with the final version. Finally, note that I have pasted in your legends at the end of the Word document and made some editorial suggestions for the legends. There is no need to write a detailed response letter - simply let me know if there are any changes that you did not make, and why. If you can turn this revision around quickly, and do not object to my changes, I would anticipate having this paper published very soon.

---

## Round 0.3 · accepted · Accept

Thank you for your second revision. This time I was able to see the supplemental figures (I'm still not sure why they did not appear previously). At this point I believe the paper is ready to publish.